# How Green Finance Affects Green Total Factor Productivity—Evidence from China

**Min Zhang [1], Chengrong Li [1], Jinshan Zhang [2,*] and Hongwei Chen [1]**

1  School of Economics and Management, Changchun University of Science and Technology, Changchun 130022, China; zm0118@cust.edu.cn (M.Z.); lcr5@mails.cust.edu.cn (C.L.); chw1123@mails.cust.edu.cn (H.C.)
2  School of Business and Management, Jilin University, Changchun 130012, China
*  Correspondence: zhangjs@jlu.edu.cn

**Abstract:** Green finance is currently a key driver of China's economic green transformation, but its impact on green total factor productivity (GTFP) mechanisms still needs further examination. Based on clarifying the theoretical mechanism of the impact of green finance on GTFP, this study uses the SBM model, which includes unexpected output, to calculate the GTFP of 30 provinces in China from 2006 to 2020. It further breaks down GTFP into green technical efficiency change (EC) and green technical change (TC) and explores in detail the internal mechanism of the impact of green finance on GTFP, as well as its channels of action from the perspectives of technology transaction activity and financial development. The study finds that green finance can significantly improve GTFP, and its impact mechanism is mainly achieved through the promotion of green technical change. Stimulating the activity of the technology transaction market and improving the level of financial development are powerful channels for green finance to improve GTFP. The study also shows that the impact of green finance is relatively robust when dividing the 30 provinces of China into coastal, inland, eastern, central, and western regions. Due to the lack of financial development and abundant natural resources, the impact of green finance is not significant in the western region. These conclusions not only provide new research perspectives and supplementary empirical evidence for understanding the impact of green finance on GTFP, but also provide useful references for further improving relevant policies and promoting China's economic development and transformation.

**Keywords:** green finance; green total factor productivity; technological change; the technical trading activity level; financial development



## 1. Introduction

Green finance is currently facilitating the transition of the economy towards more sustainable and environmentally friendly development [1–3]. For a prolonged period, the expansive economic advancement paradigm has empowered China to emancipate itself from poverty entrapment and laid the groundwork for China's economic ascension. Nevertheless, this environmentally deleterious, energy-intensive, and emission-intensive economic development paradigm has also engendered the predicament of ecological disequilibrium that has afflicted China in recent times, giving rise to recurrent scarcities of resources and predicaments of environmental contamination, impeding China's trajectory towards sustainable economic progress. It is apparent that China's verdant economic progression has emerged as an urgent concern for the government to tackle in contemporary times. The environmentally deleterious, energy-intensive, and emission-intensive economic development paradigm has engendered the predicament of ecological disequilibrium that has afflicted China in recent times, giving rise to recurrent scarcities of resources and predicaments of environmental contamination, impeding China's trajectory towards sustainable economic progress. Consequently, the 19th National Congress of the

Communist Party of China has introduced the notions of "sustainable development" and "high-quality development" in the government report, necessitating a revolutionization of production and lifestyle, a modification in growth catalysts, and a transition from a high-velocity economic growth paradigm to a high-quality economic progress paradigm [4]. The shift of the economic expansion model from "scale and speed-oriented" to "quality and efficiency-oriented" implies a shift from striving for an augmentation in GDP growth rate to striving for an augmentation in green total factor productivity. However, owing to the substantial uncertainties and expenses associated with the green metamorphosis of economic development, substantial policy and financial assistance must be provided. In this context, the establishment of a green financial system has emerged as a paramount concern for the current green economic transformation. The significance of establishing a green financial system in China was initially proposed in the "Overall Plan for Ecological Civilization System Reform", circulated by the Chinese government in 2015, and was further expounded upon in the "Guiding Opinions on Building a Green Financial System", circulated in 2016, which defined green finance as financial assistance that sustains ecological conservation, ameliorates environmental conditions, and endorses resource utilization efficiency. Specifically, there are many factors that affect the green transformation of the economy, including green innovative technologies [5], green business financing [6], and related green knowledge capital [7]. Therefore, in the process of sustainable economic development, there exists significant uncertainty and risk in financing that cannot solely rely on objective indicators of financial returns. It is necessary to provide considerable policy support and financial assistance. Compared to conventional financial services, green finance is more conducive to channeling funds to support the development of enterprises with green environmental attributes and is also a crucial source of funding for industries to enhance their green total factor productivity through green metamorphosis. Currently, the green finance system is mainly composed of green credit, green securities, green insurance, green bonds, and government support policies. Among them, green credit plays a leading role in the green finance system and is the main financing channel for relevant green enterprises [8–10]. Green total factor productivity is a pivotal indicator for measuring economic development efficacy and sustainability, signifying the efficiency of economic green transformation. Thus, accomplishing green finance to encourage the enhancement of green total factor productivity is a vital theme for China's current sustainable economic development.

In the past few decades, traditional production and consumption patterns have indeed had significant negative impacts on the environment, including carbon emissions and resource waste. As a result, there has been increased global attention towards green development, with a particular focus on green finance as a financial approach that promotes environmental sustainability. Green finance not only considers the environmental aspects of investment projects but also plays a crucial role in directing funds towards the field of environmental protection. By doing so, it stimulates green technology innovation and application, creating a positive cycle of economic growth and environmental protection. However, the specific impact mechanism and evaluation of green finance on the growth of green total factor productivity (GTFP) are still relatively limited. It is important to examine whether green finance has a promoting effect on China's GTFP improvement and to understand the internal mechanisms driving this relationship. Additionally, it is essential to investigate whether green finance can activate the technology market, stimulate the development of a more robust financial system, and steadily enhance GTFP. Furthermore, exploring the intermediary mechanisms that influence the relationship between green finance and GTFP is crucial. To address these issues, this study utilizes panel data from 30 provinces in China spanning from 2006 to 2020. The findings will hold practical significance in understanding how green finance affects the growth of total factor productivity, improving the green finance system, and supporting China's green transformation and development.

## 2. Literature Review

### 2.1. Related Research on Green Finance

As the green transformation of the economy continues to advance, the important role of green finance in economic development is increasingly being recognized by scholars. The predecessor of the concept of green finance was environmental finance. Cowan (1998) believed that environmental finance was a financial innovation behavior in which financial institutions provided funding support to achieve environmental protection [11]. With the increasing importance of financial institutions in the development of the green economy, green finance has been defined as an efficient investment and financing activity that can help promote sustainable economic development and improve environmental quality. Under the market mechanism, it can play a role in the government's control of market resource allocation and become an important means to achieve green economic transformation [12]. Chami (2002) believed that green finance may have the disadvantage of increasing enterprises' risk identification costs, but it also makes enterprises bear more social responsibilities and promotes social development [13]. Scholtens (2006) believed that green finance is an innovative move in the development of financial institutions, which can achieve coordinated development of the economy and the environment [14]. Hoti (2007) conducted risk identification research on green finance and believed that the purpose of green finance is to achieve a win–win situation for sustainable economic, environmental, and social development [15].

In the measurement of green finance, some scholars have used indicators such as industry green investment [16], green bond issuance [17], green agricultural insurance [18], and green credit [19,20] as proxy variables for green finance. However, these single indicators, although they can represent green finance to some extent, cannot comprehensively measure the development of the green finance system. Therefore, some scholars have established a green finance evaluation system to measure its comprehensive development level. For example, Chen and Chen (2021) measured the development level of green finance in China from four aspects: green credit, green securities, green investment, and green insurance, using the entropy method [21]. Based on this, Wang and Fan (2023) constructed a green finance development index from five levels, including green credit, green investment, green securities, green insurance, and green bonds, also using the entropy method, believing that it more objectively and accurately reflects the true level of green finance development [22]. Zhou et al. (2012) evaluated the degree of green finance development from the aspects of green credit, green bonds, green insurance, green investment, and green finance using a combination of time series analysis and principal component analysis, believing that they are more comprehensive and accurate [23]. From the above literature, it can be seen that when constructing a green finance development index, it is necessary to measure it from a multidimensional perspective to avoid the one-sidedness of measuring green finance with a single variable.

### 2.2. Research on Green Total Factor Productivity

The efficient utilization of scarce resources to maximize utility or output in the production process is indeed a critical concern in the field of economics. Enhancing productivity is key to achieving efficiency improvement [24]. Total factor productivity (TFP) takes into account the output efficiency of comprehensive inputs such as labor and capital and reflects the quality of economic growth [25]. Due to the unsustainable use of resources and increasing pollution emissions resulting from long-term production and human lifestyle, the issues of resource scarcity and environmental pollution have worsened over time. Resources and environmental factors have become significant constraints on economic growth. Researchers worldwide have focused on the coordination between economic growth and environmental protection, leading to the development of environmental economics theories related to sustainable economic growth and environmental protection. Although TFP is widely recognized as the long-term driver of economic growth, it only considers traditional production factors such as capital, labor, and land inputs, without

accounting for the impact of energy and environmental factors on production efficiency. Green total factor productivity (GTFP) has been developed to address this limitation. GTFP builds upon TFP and integrates resource consumption and pollution emissions into the production function framework. This enables the measurement of dynamic changes in production efficiency under resource and environmental constraints, reflecting the concept of sustainable economic development [26].

With the continuous promotion of global economic greening in recent years, domestic and foreign scholars have analyzed the influencing factors of green total factor productivity (GTFP) from various perspectives. Xie et al. (2021) studied the promotion path of GTFP from the perspective of energy consumption transformation, believing that energy consumption transformation will only have a positive impact on GTFP within an appropriate range and that excessively high or low levels of energy conversion are not conducive to improving GTFP [27]. Zhao et al. (2022) stated that the "innovation dividend" of green innovation is the key to improving GTFP and that green innovation has a significant positive impact on GTFP, while financial development, population density, and environmental regulations are significantly negatively correlated [28]. Chen et al. (2018) analyzed the factors affecting GTFP from three aspects: institutional, technological, and structural, indicating that independent research and development can promote GTFP growth more than technology introduction, while financial development, population density, and environmental regulations are significantly negatively correlated [29]. Zhong and Li (2020) believe that financial development can promote GTFP growth through innovative channels, as financial development promotes the introduction of green innovation and energy-saving technologies [30].

### 2.3. Research on Impact of Green Finance on GTFP

Green finance, as a key driver of sustainable economic development, has gained significant attention in current financial innovation. Scholars have reached a consensus on the importance of green finance in effectively promoting sustainable development [31,32]. Green finance differs from traditional financial services by focusing more on supporting the research and development of green technologies and the growth of green industries. By facilitating the application of relevant technologies and expanding the scale of green industries, it plays a vital role in driving economic transformation towards green development. While there is relatively limited research on the impact of green finance on green total factor productivity (GTFP), some studies have provided insights into this relationship. Li et al. (2023) found that green finance can enhance enterprise GTFP through energy conservation and emission reduction, particularly benefiting enterprises with weak financing constraints [33]. Su and Cheng (2023) demonstrated that green finance significantly improves industrial GTFP based on a differential method, with the largest effect observed in the mid-stage of industrialization [34]. Lu et al. (2023) highlighted the crucial role of finance in promoting innovation and industrial structure. Their study showed that China's finance technology can enhance GTFP by fostering research and development and optimizing industrial structure, but this improvement exhibits spatial heterogeneity [35]. Xu and Zhao (2023) suggested that green finance influences GTFP through technological innovation and the upgrading of industrial structure, with a more pronounced promotion effect observed in regions with higher information technology levels [36]. Liu (2023) revealed that the abundance of natural resources inhibits GTFP growth, confirming the resource curse hypothesis. However, green finance can effectively alleviate this inhibitory effect, emphasizing its important role in sustainable economic development [37]. These studies collectively indicate that green finance has the potential to enhance GTFP by promoting energy efficiency, technological innovation, and the growth of green industries. As green finance continues to expand and evolve, further research is needed to explore its comprehensive impact on GTFP and its role in achieving sustainable economic development.

However, based on the literature review presented above, it is evident that the current research pertaining to the impact of green finance on GTFP is still in its nascent stages

compared to other studies concerning the effect on GTFP. The existing research primarily concentrates on how green finance can enhance GTFP through R&D innovation and industrial structural pathways. Nonetheless, it is important to note that the principal objective of green finance is to activate green-related innovation and R&D activities, serving as an auxiliary measure for the uncertainties and risks associated with green economic transformation in the current financial development process. Thus, green finance plays a crucial role in augmenting the activity of the technology market and promoting financial development. Through the process of enhancing GTFP via green finance, gradual improvements in technology market transactions and financial development are vital for achieving stable growth of GTFP. In light of this, the present paper undertakes further decomposition of GTFP into green technology progress and green technology efficiency from the perspectives of technology market activity and financial development. It examines the impact mechanism of green finance on GTFP and presents pertinent conclusions and recommendations. The findings of this study will better facilitate China's overall sustainable economic development.

## 3. Mechanism Analysis and Hypothesis

Green total factor productivity (GTFP) is a variable that fully considers the resources and environment and can measure the sustainable development of an economic region. It is an important indicator for China to improve the quality of its development and continuously promote economic intensification and green development. Green finance is an important measure in the current financial field to promote the green transformation of the economy. It plays a crucial role in the research and application of green technologies, effectively activating the implementation of new technologies. Moreover, as a strong complement to the financial system, it improves the shortcomings of financial development in the new era. Therefore, this article mainly analyzes the impact of green finance on GTFP from the perspectives of technological market activity and the financial development level. In order to explore the internal mechanism of GTFP in detail, this article further breaks down GTFP into green technological change (TC) and green technological efficiency change (EC) in order to provide a clearer analysis of the enhancement path of GTFP through green finance.

### 3.1. Effect of Technological Market Activity

The activity level of the technology market reflects the real situation of China's technological innovation and technology transfer. The more active the technology trading market is, the greater the investment of enterprises in technology research, development, and application; the stronger the demand for technology upgrades; and the stronger the confidence of participation in technology trading. Whether it is direct investment in technology development or services or consulting contracts signed to solve technical problems, they are the most direct driving forces for economic development and the most direct manifestation of whether the green economic transformation is successful or not [38]. Green finance has released a considerable degree of new industrial landings through targeted funding support for technology-based enterprises, improving the efficiency of innovation resource allocation, and is the most direct manifestation of green finance's promotion of innovative behavior. Under the guidance of green finance policies such as market access, tax and fee reductions, and targeted funding support, low-pollution and low-energy-consuming enterprises have received more funding support. Green finance has directly helped the research and development behavior and the application of research results of green technology research and development enterprises, leading to great help for enterprises in the introduction and application of green technology and promoting the progress of green technology (TC), achieving industrial optimization and upgrades. This direct assistance to the development of the technology market is also conducive to improving energy efficiency [39]. Therefore, this article proposes the following hypothesis:

**Hypothesis 1.** *Green finance can promote the enhancement of GTFP by increasing the activity of the technology market, and it is achieved through promoting TC.*

*3.2. The Effect of Financial Development Level*

The development of finance plays an important supportive role in economic development, as it can promote the efficient allocation of funds, risk diversification, and innovation [40]. Furthermore, financial development can facilitate sustainable economic growth through mechanisms such as capital deepening and technological innovation [41]. The financial system, by providing diverse financial products and services, helps facilitate the flow of funds for relevant enterprises, reducing uncertainty in economic operations and enhancing economic stability. The development of green finance is closely intertwined with the level of financial development. The higher the level of financial development, the stronger the support for green finance policies. Higher levels can provide greater financial support and encourage enterprises and institutions to engage in environmentally friendly investments and the development of green projects [42]. As the green finance system continues to evolve, it can screen and manage environmentally friendly industries and projects, thereby reducing environmental risks faced by the financial system and investors. This helps protect the interests of financial institutions and investors while minimizing adverse impacts on the environment, thus strengthening the financial system. The more robust the development of the financial system, the more effectively it can drive economic transformation and sustainable development. Through financial support and market incentives, green technology research and application receive strong support, promoting the transition of the economy towards a green and low-carbon direction [43,44]. The active introduction of green finance concepts and practices by financial institutions during their development contributes to achieving sustainable economic development and improving quality. Therefore, this article proposes the following hypotheses:

**Hypothesis 2.** *Green finance can promote the enhancement of GTFP by improving financial development, and it is achieved through promoting TC.*

**4. Model Specification and Variable Description**

*4.1. Model Setting*

(1)　Calculation and decomposition of GTFP.

Data envelopment analysis (DEA) is a method in operations research and the study of economic production boundaries, commonly used to measure the production efficiency of decision-making units. This method can calculate the efficiency of multiple inputs and outputs without the need to specify a specific functional form, thus avoiding structural biases caused by the misspecification of production functions in traditional accounting methods. Given that the traditional DEA model has not been able to effectively address the issue of slack variables in efficiency evaluation and that it cannot distinguish well between decision-making units with an efficiency of one, this paper adopts the SBM model proposed by Tone [45], which considers unexpected outputs. This model can comprehensively consider the relationship between inputs, outputs, and pollution outputs and can effectively solve the slack problem in efficiency evaluation.

In this study, we construct a production decision-making unit (DMU) for Chinese provinces to establish an optimal production technology frontier. By combining the non-radial SBM model with an undesirable output, we construct a distance function for the Malmquist index. Using the Malmquist index decomposition method, we break down the growth rate of green total factor productivity (GTFP) into green technology change (TC) and green technology efficiency change (EC).

$$GTFP_i^{t,t+1} = TC_i^{t,t+1} \times EC_i^{t,t+1} \tag{1}$$

$TC_i^{t,t+1}$ indicates the progress of green technology in the $i$ province during the period from $t$ to $t+1$, specifically the movement of the technological frontier. Meanwhile, $EC_i^{t,t+1}$ represents the changes in the efficiency of green technology in the $i$ province during the period from $t$ to $t+1$.

(2)　The impact model of green finance on GTFP.

In order to examine the impact of green finance on GTFP, this paper constructs an econometric model with green finance as the core explanatory variable and GTFP as the explained variable.

$$GTFP_{it} = \beta_0 + \beta_1 GF_{it} + \beta_2 x_{it} + \theta_i + \mu_t + \varepsilon_{it} \tag{2}$$

In Equation (2), $i$ represents the provinces of China, $t$ represents the corresponding year, $\beta_0$ represents the constant term, $\beta_1$ and $\beta_2$ represent the coefficients of the core explanatory variables and control variables, $\theta_i$ represents the unobservable individual fixed effects, $\mu_t$ represents the unobservable time fixed effects, and $\varepsilon_{it}$ represents the random error term. $GTFP$ represents the green total factor productivity, which is the research object of this paper; $GF$ represents the core explanatory variable of green finance; and $x_{it}$ represents the corresponding control variable.

In order to verify the mechanism of influence between GTFP and green finance, this paper employs a mediation model and establishes the following model to describe its mechanism of influence:

$$GTFP_{it} = cGF + e_1 \tag{3}$$

$$M_{it} = aGF_{it} + e_2 \tag{4}$$

$$GTFP_{it} = c\prime GF_{it} + bM_{it} + e_3 \tag{5}$$

In the equation, $M$ represents the mediating variable, with the same meaning as the variables in Model (2). In Equation (3), $c.$ denotes the overall effect of green finance on GTFP, while in Equation (4), $a$ represents the impact of the mediating variable on GTFP after controlling for the influence of green finance. $ab$ signifies the effect of green finance on GTFP through the mediating variable.

### 4.2. Variable Description and Data Explanation

(1)　The explained variable: green total factor productivity.

The green total factor productivity not only emphasizes efficiency, but also pays attention to the quality of economic development, which can explore the driving force of sustainable development at a deeper level and measure the quality of economic development more accurately and comprehensively. When calculating the green total factor productivity, natural resource consumption and pollutants that affect environmental degradation and climate change are taken into consideration. Therefore, this paper uses the SBM and Malmquist index models to calculate the GTFP index of 30 provinces in China from 2006 to 2020. Specifically, in terms of labor input, the number of employees at the end of each year in each province is used as the labor input indicator, with a unit of ten thousand people. In terms of capital input, the perpetual inventory method is used to calculate the capital stock at the end of each year in each province, with the year 2005 as the base period and a unit of hundred million yuan. In terms of energy input, the energy consumption of each province is used as the energy input indicator, with a unit of ten thousand tons of standard coal. The entropy weight method is used to calculate the comprehensive values of industrial wastewater, carbon dioxide, and industrial sulfur dioxide emissions in each province as the indicator of harmful pollutant output. With the year 2005 as the base period, the unit is hundred million yuan. The energy input indicator is also the energy consumption of each province, with a unit of ten thousand tons of standard coal. The comprehensive values of industrial wastewater, carbon dioxide, and industrial

sulfur dioxide emissions in each province are calculated using the entropy weight method as the indicator of harmful pollutant output. Please refer to Table 1 below for details.

**Table 1.** Main variables and calculation methods of GTFP input–output analysis.

| Category of Indicators | Indicator Name | Indicator Content |
| --- | --- | --- |
| Input indicators. | Capital input | The fixed-asset capital stock of each province calculated using the perpetual inventory method. |
| | Labor input | End-of-year employed population in each province. |
| | Energy input | Energy consumption (in 10,000 tons of standard coal) in each province. |
| Expected output indicators. | Economic output | The local gross domestic product calculated at constant prices in 2005. |
| Unexpected output indicators. | Environmental Pollution Index | The comprehensive calculation of industrial wastewater, carbon dioxide, and industrial sulfur dioxide emissions for each province, conducted using the entropy weight method. |

(2) Core explanatory variable—green finance.

The original intention of establishing green finance is to make up for the shortcomings in financial support for sustainable economic and social development during the process of financial system development. From the current relevant research, there is no unified consensus on the measurement of green finance. Based on the actual service scope of green finance, the current scope of green finance services in China mainly includes green credit, green investment, green insurance, and government policy support. Therefore, as shown in Table 2, this paper selects these four aspects to measure green finance in China and calculates the comprehensive score of green finance from 2006 to 2020 using the entropy method. Based on the actual service scope of green finance, the current scope of green finance services in China mainly includes green credit, green investment, green insurance, and government policy support. Therefore, as shown in Table 2, this paper selects these four aspects to measure green finance in China and calculates the comprehensive score of green finance from 2006 to 2020 using the entropy method.

**Table 2.** System of green finance indicators.

| Primary Indicator | Indicator | Indicator Description |
| --- | --- | --- |
| Green credit | Proportion of interest expenditure in high-energy-consuming industries | Interest expenditure on the six major high-energy-consuming industrial sectors as a proportion of total industrial interest expenditure<br>Green investment |
| Green investment. | Proportion of environmental pollution control investment to GDP | Environmental pollution-control investment as a percentage of GDP |
| Green insurance | Agricultural insurance intensity | Agricultural insurance income as a proportion of total agricultural output value |
| Government support | Proportion of fiscal expenditure on environmental protection | Expenditure on fiscal environmental protection as a proportion of total fiscal budget expenditure |

(3) Mediating variables.

Based on the previous analysis, the following variables were selected as the mediator variables in this paper:

① Technology Market Intensity (TMI): Technology trading, focused on providing innovation and technology services for social development, plays a crucial role that cannot be replaced by other market factors. It is centered on innovation and technology services, with the goal of diffusing knowledge and transferring technology to truly transform

technological advantages into economic advantages. Enhancing the intensity of the technology market can promote sustainable economic and social development and improve the quality of economic development. This article measures the level of local technology market activity by selecting the proportion of the annual technology market turnover to the local GDP in each province of China.

② Level of financial development (LFD): Financial development focuses on improving the efficiency of fund allocation in the operation of the social economy, providing financial support for the development of enterprises and industries, promoting investment and innovation, and resisting uncertainty in the process of economic development. A sound financial system can promote economic stability and sustainable growth. Green finance is an innovative service in the process of financial development, and the stronger the financial development, the greater the support provided for green finance. This article selects the proportion of loan balance to local GDP in each province of China to measure financial development.

(4) Control variables.

The selected control variables in this article include population size (POP), dependency on foreign trade (DFT), research and development (R&D), total retail sales of consumer goods (TRSCG), and foreign direct investment (FDI). The specific details are as follows:

① Population size (POP): The size of the population has direct and indirect impacts on economic development. A larger population can provide more labor resources, promote economic growth, and expand the output scale. However, rapid population growth can also lead to excessive resource consumption and exacerbate environmental pollution. In this article, the annual average population of each province is used to represent the population size.

② Dependency on foreign trade (DFT): The increase in the degree of dependence on foreign trade can bring benefits such as foreign exchange income, technology transfer, and knowledge renewal, which can maintain a good international trade environment and competitiveness, thereby improving GTFP. However, excessive reliance on the export of certain resources or industries may lead to excessive resource consumption and increased environmental pressure, which can affect GTFP. In this article, the degree of trade dependence of each province in a given year is represented by the ratio of the number of exports to the local GDP.

③ Research and development (R&D): Research and development (R&D) investments can directly enhance innovation and technological progress, thereby affecting gross total factor productivity (GTFP). However, due to the phenomenon of carbon lock-in, R&D activities centered around fossil fuels may crowd out the development of green technologies, thereby affecting the growth of GTFP. In this paper, the proportion of local fiscal scientific expenditure to local GDP in each province is used to represent R&D.

④ Total retail sales of social consumer goods (TRSCG): The total retail sales of consumer goods in a society reflect the local economic development status and the level of consumption activity of individuals and households. A higher total retail sales of consumer goods can stimulate economic growth, but it can also lead to excessive resource consumption and increased environmental pressure, thereby affecting GTFP. In this study, the total retail sales of social consumer goods in each province in the given year are used to represent this indicator.

⑤ Foreign direct investment (FDI): FDI typically brings advantages such as advanced production technology, capital, management expertise, and market access. However, it can also lead to some multinational corporations pursuing short-term economic gains while neglecting the importance of environmental protection. This may result in excessive resource exploitation and the transfer of pollution to the host country. The active guidance and proper regulation of foreign investment play a crucial role

in enhancing GTFP. In this study, the amount of foreign direct investment in each province is used to represent this indicator.

(5) Data sources and descriptive statistics.

The empirical data used in this article consist of panel data from 30 provincial-level administrative regions in China from 2006 to 2020. The original data of the variables involved in this article are from the "China Statistical Yearbook", the "China Energy Statistical Yearbook", the "China Environmental Statistical Yearbook", and the official website of the National Bureau of Statistics (due to incomplete data, Tibet Autonomous Region, Taiwan Province of China, Hong Kong, and Macao Special Administrative Region were excluded). The descriptive statistics of the specific variables are shown in Table 3.

**Table 3.** Descriptive statistics of variables.

|  | Variable | Obs | Mean | Std. dev. | Min | Max |
|---|---|---|---|---|---|---|
| Dependent variable. | GTFP | 450 | 0.924 | 0.239 | 0.475 | 2.178 |
|  | EC | 450 | 0.682 | 0.106 | 0.174 | 1.137 |
|  | TC | 450 | 0.936 | 0.192 | 0.644 | 2.437 |
| Core variable. | GF | 450 | 0.149 | 0.081 | 0.049 | 0.609 |
| Mediating variable. | TMI | 450 | 0.013 | 0.024 | 0.000 | 0.175 |
|  | LFD | 450 | 2.996 | 1.143 | 1.288 | 8.131 |
| Control variables. | POP | 450 | 6.003 | 0.763 | 3.789 | 7.667 |
|  | DFT | 450 | 0.284 | 0.319 | 0.007 | 1.708 |
|  | RD | 450 | 0.010 | 0.006 | 0.000 | 0.032 |
|  | TRSCG | 450 | 8.524 | 1.066 | 5.213 | 10.668 |
|  | FDI | 450 | 14.493 | 1.676 | 7.990 | 16.932 |

## 5. Analysis of Empirical Results

The annual trend of China's GTFP and its decomposition components is shown in Figure 1. Overall, both GTFP and its decomposition components show an upward trend between 2006 and 2020. From Figure 1, it can be observed that EC remained relatively stable during the study period, but shows a downward trend after 2018, which may be related to the economic crisis in that year. On the other hand, the trend of GTFP is not significantly influenced by EC but is closely related to the trend of TC. It can be inferred that TC plays a dominant role in the growth of GTFP. Additionally, GTFP also shows a downward trend after 2018, which may be associated with the economic crisis in that year. Therefore, it can be concluded that the trend of GTFP is not heavily influenced by EC, and its growth is primarily driven by TC.

### 5.1. Regression of the Impact Mechanism of Green Finance on GTFP

The regression results of the impact of green finance on GTFP are shown in Table 4. This study adopts a progressive regression processing method. Firstly, the univariate regression of the core explanatory variable is controlled by the province and year fixed effects. Secondly, the fixed effects are controlled separately, and relevant control variables are included for testing. Finally, the regression test is conducted again while simultaneously controlling for the province and year fixed effects. The results show that the coefficient of the impact of green finance on China's green total factor productivity is significantly positive at the 1% level, and the model results are highly robust. This indicates that the development of green finance has indeed significantly promoted the improvement of China's total factor productivity. In terms of the service scope of green finance, by carrying out business expansions of financial tools such as green investment, green credit, and green insurance, capital investment is increased for relevant green environmental protection enterprises, which in turn increases the research and development of green technology and the introduction and absorption of related technologies. Moreover, it improves the allocation of social resources and reduces the risks of implementing relevant clean development projects.

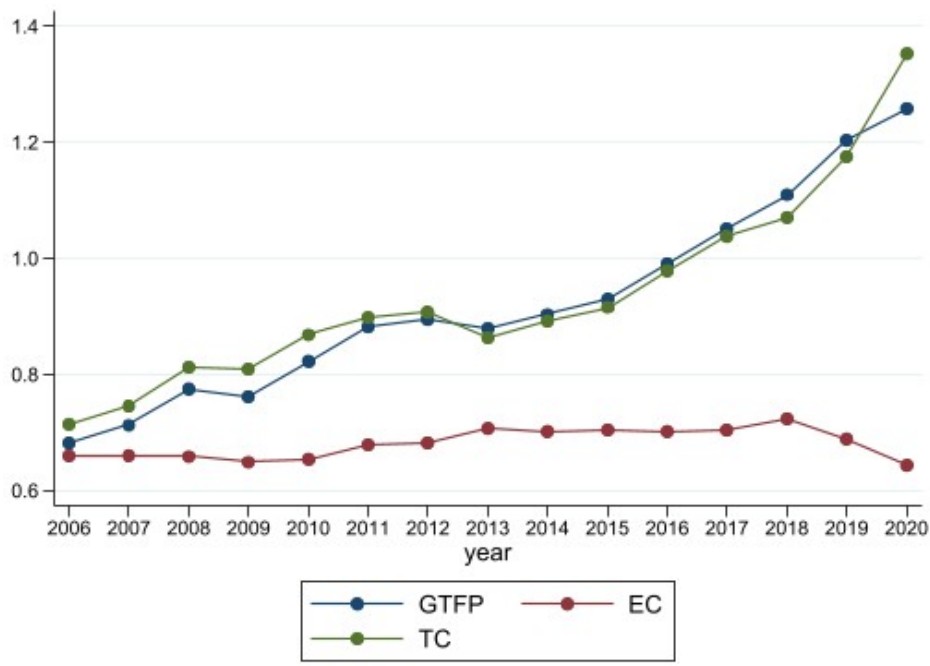

**Figure 1.** Trend chart of GTFP and its decomposition items.

**Table 4.** Regression analysis of the impact of green finance on GTFP.

| Variables | GTFP | GTFP | GTFP | GTFP |
|---|---|---|---|---|
| GF | 1.360 *** | 2.431 *** | 0.692 *** | 1.683 *** |
|  | (0.253) | (0.255) | (0.117) | (0.264) |
| POP |  | −0.191 *** | −0.238 *** | −0.031 |
|  |  | (0.062) | (0.041) | (0.069) |
| DFT |  | 0.042 | 0.011 | 0.109 * |
|  |  | (0.064) | (0.030) | (0.065) |
| R&D |  | −13.468 *** | 6.856 *** | −11.808 *** |
|  |  | (3.088) | (2.003) | (2.939) |
| TRSCG |  | 0.167 *** | 0.138 *** | −0.032 |
|  |  | (0.023) | (0.035) | (0.039) |
| FDI |  | −0.021 *** | −0.007 | 0.000 |
|  |  | (0.004) | (0.009) | (0.006) |
| Constant |  | 0.205 | 1.017 *** | 0.691 |
|  |  | (0.320) | (0.155) | (0.423) |
| Observations | 450 | 450 | 450 | 450 |
| Number of ids | 30 | 30 | 30 | 30 |
| R-squared | 0.803 | 0.785 | 0.589 | 0.817 |
| Province FE | YES | YES | NO | YES |
| Year FE | YES | No | YES | YES |

Standard errors in parentheses, *** $p < 0.01$, * $p < 0.1$.

China has undergone an extensive and protracted period of economic development, resulting in rapid economic growth, albeit at the expense of environmental degradation and resource squandering. The shift from extensive to intensive sustainable developmental models necessitates the augmentation of GTFP. Drawing upon prior research and analysis [26,46], it has been established that green technological progress constitutes the pivotal driving force behind the enhancement of GTFP. As a consequence, this paper dissects GTFP into two components, namely, green technological efficiency and green technological progress, so as to delve deeper into the fundamental catalysts that underpin the influence of green finance on GTFP. As depicted in Table 5, the regression analysis pertaining to the constituent elements of GTFP, specifically, EC and TC, reveals that green finance exerts no statistically significant impact on green technological efficiency, while demonstrating

a substantial positive effect on green technological progress at a remarkable 1% significance level. This serves to signify that the existing green finance framework in China can significantly uplift the echelons of green technology and fortify GTFP through the facilitation of green technological progress, thus corroborating previous findings that TC constitutes the primary impetus behind GTFP. Green finance exerts a constructive impetus on the financing of interconnected green industries and the research and development of sustainable projects, thereby stimulating the pursuit, adoption, and application of green technology within the industrial realm, ultimately precipitating a momentous amelioration in the realm of green technological progress.

**Table 5.** The impact of green finance on EC and TC.

| Variables | EC | TC |
|:---:|:---:|:---:|
| GF | 0.261 | 0.719 *** |
| | (0.166) | (0.193) |
| POP | 0.072 * | −0.262 *** |
| | (0.044) | (0.050) |
| DFT | 0.172 *** | −0.143 *** |
| | (0.041) | (0.048) |
| R&D | −3.009 | −5.382 ** |
| | (1.850) | (2.144) |
| TRSCG | −0.016 | 0.042 |
| | (0.024) | (0.028) |
| FDI | 0.004 | −0.004 |
| | (0.004) | (0.005) |
| Constant | 0.108 | 2.052 *** |
| | (0.266) | (0.309) |
| Observations | 450 | 450 |
| Number of id | 30 | 30 |
| R-squared | 0.628 | 0.849 |
| Province FE | YES | YES |
| Year FE | YES | YES |

Standard errors in parentheses, *** $p < 0.01$, ** $p < 0.05$, * $p < 0.1$.

### 5.2. Testing the Mediating Effect of Green Finance on GTFP and TC

Following the empirical procedures of the preceding benchmark regression, it can be deduced that green finance wields a positive influence on GTFP at a remarkable 1% significance level. Moreover, upon conducting a regression analysis of the constituent elements of GTFP, it was discerned that the underlying mechanism through which green finance impacts GTFP is achieved by fostering green technological progress. Consequently, this investigation scrutinizes the mediating effects of green finance on GTFP and TC through two avenues: invigorating the market price activity of technology and heightening the level of financial development. The outcomes of the mediation analysis are delineated in Table 6.

From the vantage point of technological market activity, the impact coefficient of green finance on technological market activity is notably positive, signifying that green finance can effectively kindle an escalation in the transaction volume within the technological market. By means of venture capital, entrepreneurial support, as well as green bonds, green loans, and government and market guidance, green finance has contributed to mitigating the financing costs of enterprises and research institutions, thus enabling innovative enterprises to achieve breakthroughs in the sphere of technological research, development, and commercialization. Ergo, green finance directly stimulates the activity of the technological trading market, rendering it more accessible for related industries to delve into, introduce, and commercialize technology. In light of the results of the intermediary effect, the impact coefficients of technological trading activity on GTFP and TC are significantly positive, and both withstand the Sobel test, thereby indicating the establishment of the intermediary effect. Hence, green finance can propel the establishment of the GTFP framework by invigorating the technological market activity.

**Table 6.** The examination results of the intermediary effect of green finance on GTFP.

| Variables | TMI | GTFP | TC | LFD | GTFP | TC |
|---|---|---|---|---|---|---|
| GF | 0.278 *** | 0.262 | 0.348 ** | 9.689 *** | 0.601 *** | 0.218 * |
| | (0.011) | (0.182) | (0.141) | (0.586) | (0.148) | (0.111) |
| TMI | | 1.891 *** | 0.721 * | | | |
| | | (0.507) | (0.393) | | | |
| LFD | | | | | 0.019 ** | 0.034 *** |
| | | | | | (0.009) | (0.007) |
| POP | 0.008 *** | −0.367 *** | −0.265 *** | −0.677 *** | −0.338 *** | −0.236 *** |
| | (0.003) | (0.028) | (0.022) | (0.143) | (0.029) | (0.022) |
| DFT | 0.006 ** | −0.017 | 0.021 | 1.345 *** | −0.031 | −0.020 |
| | (0.003) | (0.029) | (0.023) | (0.148) | (0.032) | (0.024) |
| R&D | −0.708 *** | 7.178 *** | −0.715 | −57.750 *** | 6.948 *** | 0.744 |
| | (0.187) | (2.028) | (1.571) | (10.167) | (2.090) | (1.567) |
| TRSCG | −0.011 *** | 0.256 *** | 0.200 *** | 0.228 ** | 0.231 *** | 0.184 *** |
| | (0.002) | (0.023) | (0.018) | (0.116) | (0.0230) | (0.017) |
| FDI | −0.001 * | −0.030 *** | −0.039 *** | −0.053 * | −0.031 *** | −0.038 *** |
| | (0.000) | (0.005) | (0.004) | (0.027) | (0.005) | (0.004) |
| Constant | 0.062 *** | 1.311 *** | 1.395 *** | 3.072 *** | 1.370 *** | 1.335 *** |
| | (0.011) | (0.121) | (0.093) | (0.593) | (0.121) | (0.091) |
| Observations | 450 | 450 | 450 | 450 | 450 | 450 |
| Number of ids | 30 | 30 | 30 | 30 | 30 | 30 |
| R-squared | 0.715 | 0.571 | 0.600 | 0.512 | 0.561 | 0.617 |
| Sobel test | | 0.526 *** | 0.200 ** | | 0.186 ** | 0.200 ** |
| | | (0.142) | (0.11) | | (0.092) | (0.11) |

Standard errors in parentheses, *** $p < 0.01$, ** $p < 0.05$, * $p < 0.1$.

From the perspective of financial development, the impact coefficient of green finance on financial development is significantly positive, which implies that enhancing the level of green finance can foster financial development. Green finance, with environmental sustainability as its objective, serves as a financial development model for the green transformation of the economy. It is an effective supplement to the insufficient targeted support of the financial system for the green industry during China's transition from extensive to intensive economic development. The screening and management of new environmental protection technology projects in the operation of green finance also decrease the financing risk of the financial system. With the development of green finance, the landing of targeted support industries and the successful commercialization of associated projects have successfully stimulated the growth of new local industries. Additionally, green finance enhances the allocation efficiency of social financing, strengthens the health of the financial system, improves the quality of financial development, and contributes to sustainable economic development. From the perspective of intermediary effects, the impact of financial development on GTFP and TC is notably positive, and both withstand the Sobel test, thereby indicating the establishment of the intermediary effect. As the level of financial development improves, various types of capital become more concentrated, rendering various investment and financing projects more accessible, which aids in enhancing the scale of local economic development, attaining economies of scale, and boosting the efficiency of resource utilization. Therefore, green finance can advance the establishment of the GTFP framework by elevating the level of financial development.

*5.3. Robustness Test*

In order to ensure the accuracy of the regression results, robustness tests are conducted on the econometric model in this study. The method of grouped regression is employed, dividing the 30 provinces of China into inland provinces, coastal provinces, an eastern region, a central region, and a western region. As shown in Table 7, except for the western region, the remaining groups have a significant positive impact on GTFP with regards to green finance, and the temporal conclusion of this study remains unchanged. The

western region is limited by its underdeveloped financial system and strong dependence on resources for economic development due to its abundant resource endowment, which has hindered the development of green finance and resulted in a "resource curse" effect on the green transformation of the economy. This conclusion is consistent with some of the viewpoints presented in the study conducted by Lin and Zhou [47].

**Table 7.** Robustness test.

|  | **Inland Provinces** | **Coastal Provinces** | **Eastern Provinces** | **Central Provinces** | **Western Provinces** |
|---|---|---|---|---|---|
| GF | 1.699 *** | 1.652 ** | 1.336 *** | 3.367 *** | −0.595 |
|  | (0.316) | (0.660) | (0.400) | (1.081) | (0.790) |
| Control variables | Control | Control | Control | Control | Control |
| Constant | −0.219 | 3.798 *** | 3.892 *** | −0.533 | 0.129 |
|  | (0.486) | (0.815) | (1.057) | (0.784) | (0.704) |
| Observations | 450 | 450 | 450 | 450 | 450 |
| Number of ids | 30 | 30 | 30 | 30 | 30 |
| R-squared | 0.862 | 0.697 | 0.780 | 0.874 | 0.910 |
| Province FE | YES | YES | YES | YES | YES |
| Year FE | YES | YES | YES | YES | YES |

Standard errors in parentheses, *** $p < 0.01$, ** $p < 0.05$.

## 6. Conclusions and Policy Recommendations

Comprehending the impact mechanism of green finance on GTFP and advancing more targeted policies of green finance in China bear substantial theoretical and pragmatic significance in attaining a verdant transformation of the economic development model. Presently, research in the domain of green finance is relatively circumscribed in evaluating its impact and has not diligently concentrated on the intrinsic impact mechanism of GTFP. This study explores the pathway of green finance in fortifying GTFP and additionally analyzes its internal mechanism for bolstering GTFP.

Research shows that: ① Green finance can significantly improve the growth of green total factor productivity (GTFP) and effectively promote China's green transformation and development. By breaking down GTFP into green technology efficiency and green technology progress, further exploration of the internal impact mechanism of green finance on GTFP reveals that green finance enhances GTFP by promoting green technology progress. ② Activating the technology market and improving the level of financial development are effective channels for green finance to enhance GTFP. Green finance stimulates the market's research and development and the commercial application of green-related industries, significantly promoting the commercialization of new industries and thus improving GTFP. ③ Robustness analysis shows that the development of green finance has a stable impact on GTFP. By dividing 30 provinces in China into coastal, inland, eastern, central, and western regions, the regression results show that green finance has a positive effect on GTFP. However, the impact of green finance on GTFP in the western region is weaker due to the lack of financial development and the strong economic dependence on resources.

The research conclusions of this article can provide the following policy inspirations for relevant departments in China: ① Vigorously propelling the advancement of verdant finance and harnessing the full potential of its policy ramifications serve as potent instruments for fostering the metamorphosis of China's economic progression. From the vantage points of verdant securities, verdant credit, verdant investment, verdant insurance, and governmental policy guidance, it becomes imperative to refine the pertinent laws and regulations governing verdant finance, enhance the information disclosure framework therein, and fully harness the directive influence of the government. The successful practices witnessed in regions exhibiting commendable performances in verdant finance ought to be disseminated nationwide, augmenting the echelon of verdant finance development and thus effectively elevating China's technological prowess and stimulating sustainable economic growth. The government should wholeheartedly exploit its directive role in

refining the information disclosure framework of verdant finance and disseminating the successful practices observed in regions showcasing commendable performances in verdant finance nationwide, thus bolstering the echelon of verdant finance development, which can efficaciously enhance China's level of technological progress and propel sustainable economic advancement. ② In order to augment the technological market's trading system and fortify pertinent supportive policies, China ought to proactively foster the advancement and assimilation of eco-centric technologies, along with the commercialization of eco-friendly endeavors. Such endeavors can efficaciously propel the swift progression of China's eco-related sectors and amplify the scale effect of nascent industries. Consequently, it behooves the Chinese government to refine the technology market's trading system, heighten the efficacy of information alignment within the technology market, and earnestly steer the inter-firm exchange of technology. ③ In order to actively improve China's financial development level, financial institutions should enhance their ability to develop green finance business, actively respond to national policies, and enhance the business scale of green finance. When formulating green finance policies and financial support policies, the impact of external environments in various regions should be fully considered, and targeted investment and financing should be carried out to truly play the role of green finance policies in promoting high-quality economic development in local areas.

This article is based on a macro perspective to study the impact of China's green finance and GTFP. However, in actual economic operation, the most direct impact of green finance is on enterprises. Therefore, in future research, we will adopt a macro–micro combined perspective to analyze the impact mechanism of green finance.

**Author Contributions:** Conceptualization, M.Z. and C.L.; methodology, M.Z.; software, M.Z.; validation, M.Z. and C.L.; formal analysis, M.Z.; investigation, J.Z.; resources, M.Z.; data curation, M.Z.; writing—original draft preparation, M.Z.; writing—review and editing, M.Z.; visualization, M.Z and H.C.; supervision, J.Z.; project administration, J.Z.; funding acquisition, J.Z. All authors have read and agreed to the published version of the manuscript.

**Funding:** This research was funded by Humanities and Social Science Fund of Ministry of Education (20YJC630209); Changchun University of Science and Technology Base Support Special Project (KYC-JDY-2021-01).

**Institutional Review Board Statement:** Not applicable.

**Informed Consent Statement:** Not applicable.

**Data Availability Statement:** For the purpose of further research, this article does not provide data at this time.

**Acknowledgments:** The authors are grateful to the editor and the anonymous reviewers of this paper.

**Conflicts of Interest:** The authors declare no conflicts of interest.

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
