# Peer review of "How Green Finance Affects Green Total Factor Productivity—Evidence from China"

_sustainability, doi:10.3390/su16010270_

Round 1

Reviewer 1 Report

Comments and Suggestions for Authors

The paper contains a comprehensive analysis of the impact of green finance on green total factor productivity (GTFP) in China, providing valuable insights into the mechanisms and effects of green finance on economic transformation. Here are some recommendations to strengthen and enhance the paper:

1. Data and Analysis:

- Contextualization of Data: Explain the relevance and significance of the chosen variables and their impact on GTFP in the Chinese context. This contextualization can provide a clearer understanding of the implications of the findings.

2. Conclusion and Recommendations:

- Concise Summary: Condense the conclusion section to emphasize the most crucial findings and their implications for policymakers and researchers.

- Policy Recommendations: Strengthen the policy recommendations by providing a more concise and actionable set of suggestions for relevant departments in China. Clear, actionable recommendations are more likely to be implemented.

3. Enhanced Explanation:

- Elaborate on Methodologies: While the article delves into methodologies used for calculations, consider providing a clearer explanation for readers less familiar with those specific models (e.g., SBM model, Malmquist index), ensuring they can grasp the applied techniques.

By addressing these points, the paper can become more accessible, engaging, and impactful for a wider audience interested in the intersection of green finance and economic productivity in China.

Reviewer 2 Report

Comments and Suggestions for Authors

The conclusions are interesting but are not fully justified based on the study conducted in the reviewed article. The following points in the reviewed article require some revision.

1. Five variables have a coefficient of variation (not calculated in the reviewed article) that is quite high. Their use in the model becomes problematic for the formulated conclusions of the study for all regions of China. ( p.10, Table 3).

The two variables, TMI and DFT have a standard deviation value excessively higher than the mean value. The TMI variable has a standard deviation value of almost twice the average value. The TMI variable's variation coefficient (not calculated in the article) is higher than 1. The maximum and minimum values confirm these two variables' very large range.

The next variables, RD and GF have a coefficient of variation values (not calculated in the article) of 0.6 and more than 0.5, respectively. The variable LFD has a coefficient of variation value (not calculated in the article) of almost 0.4.

So should the TMI and DFT variables be used in the model? At this stage of the calculations presented: no.

Should the RD and GF variables be used in the model? At this stage of the presented calculations: in principle: no. In addition, the GF variable is treated, as a core variable.

Not knowing the data, but wanting to help the authors, I suggest not including extreme values, especially 0 or close to zero, in the calculations. This should improve the relationship of the standard deviation to the mean for all five variables, especially the two TMI and DFT.

Based on the calculation of the conclusions for the Western China region throughout the reviewed article, and especially on p. 15 (5.3 Robustness Test and Table 7), I first propose not to include data from the Western region. Not taking these data into account will improve the statistical quality of the study, and statistical justification will be obtained for the conclusions drawn. If the value of the coefficient of variation does not improve, I suggest removing the next extreme value of 0 or close to zero from another region. Of course, please note that the survey does not consider the Western region or any other region of China.

2. It is unreasonable to treat the FDI variable as confirmation of the "pollution halo" and "pollution refuge" hypotheses. Not all FDIs are in dirty industries, and not all of them support such two hypotheses. I propose for confirmation of these two hypotheses, to use FDI data only in dirty industries.

3. The reviewed article reveals the problem of using data that have absolute and relative value. The FDI variable has an absolute (absolute) value, and the DFT variable has a relative value. The difference between the two is very large based on simple calculations in Table 3 (p. 10). Just compare the value of the coefficient of variation (not calculated in the article) for the FDI and DFT variables. In such studies, data with relative values are best for formulating conclusions. I propose that the FDI variable in dirty industries be turned into a relative variable. Likewise, I propose that the average annual population (POP variable) be turned into a variable with a relative value as well.

4. In the reviewed article, it is worth making a model of the research procedure stages and the assumed positive and negative impact of variables on other variables. E.g. in the selected control variables is the variable TRSCG (total retail sales of social consumer goods). The authors report that 'higher total consumption stimulates economic growth'. However, based on, for example, the FDI and other variables, one might think that green finance and related sustainable development are the most important in the article. Thus, stimulating economic growth alone should not be the focus of the study.

5. The authors should clearly, unambiguously, and succinctly (e.g., as in Table 2, p. 9) define very important concepts, especially for understanding the variables, e.g., how they understand economic growth versus sustainable development and social consumer goods. Indeed, one may think that there is a need to shift away from the traditional understanding of economic growth to green finance and sustainable development. However, in this context, the question arises: why should social consumer goods only stimulate economic growth and not sustainable development? The answer to this question: depends on what one means by economic growth, sustainable development and social consumer goods.

6. The reviewed article needs editorial correction. There are repetitions of the same sentences, e.g.

‘The western region is limited by its underdeveloped financial system and strong dependence on resources for economic development due to its abundant resource endowment, which has hindered the development of green finance and resulted in a "resource curse" effect on the green transformation of the economy’ and ‘This study explores the path of green finance in enhancing GTFP, and further analyses its internal mechanism for enhancing GTFP’. (p.15 and p. 16, first paragraph)

The reviewed article, once revised, is worth publishing. The reviewed article will become interesting, and its conclusions will become more important for green finance and sustainable development in China and other countries.

Comments on the Quality of English Language

Reviewer 3 Report

Comments and Suggestions for Authors

The theme of this article is the impact of green finance on the Chinese economy, with special attention to its impact mechanism on green total factor productivity (GTFP). The conclusion of the article includes that green finance can significantly improve GTFP, and the impact of green finance is relatively weak in western China due to insufficient financial development and abundant natural resources. The structure of the article is very comprehensive, but there are still details that need to be polished.

1. In the introductory section, it is imperative to incorporate relevant citations from authoritative sources within the field and use actual data to support research viewpoints to enhances the credibility of the study and bolsters the contextual foundation of the research.

2. In section 2.3 of the article, previous research on the impact of green finance on GTFP can be organized into a table, clarifying the research methods they adopted and reasonably introducing subsequent research methods.

3. The article lacks a discussion section, and the results obtained from the research analysis need to be supported by previous literature.

4. What are the innovative points and shortcomings of the article. These two parts should be added to the study.

5. The format of article tables should be as consistent as possible. For example, Table 1, Table 2, Table 4, and Table 7. Please refer to the format required by the journal to modify them.

6. The ordinate of Figure 1 requires a coordinate name.

Reviewer 4 Report

Comments and Suggestions for Authors

Review of the article “How green finance affects total factor productivity – Evidence from China

In my review I would like to turn authors’ attention to four specific issues.

Firstly, one of the biggest disadvantage of the article is a lack of extensive discussion at the beginning of the article. In the first two pages the authors did not refer to any papers. Almost all references appear in second section of the paper, almost none in section 3-5. Moreover, the authors did not confront applied methodology (model setting), as well as obtained results, with other authors.

Secondly, in my opinion, the issue of green bonds has not been sufficiently highlighted. In this regard, it is suggested to refer to the following articles, which show cause-and-effect relationships between such instruments and other variables representing other asset classes.

Ferrer, R.; Shahzad, S.J.H.; Soriano, P. Are green bonds a different asset class? Evidence from time-frequency connectedness analysis. J. Clean. Prod. 2021, 292, 125988

Ferrer, R.; Benítez, R.; Bolós, V.J. Interdependence between Green Financial Instruments and Major Conventional Assets: A Wavelet-Based Network Analysis. Mathematics 2021, 9, 900

Orzechowski, A.; Bombol, M. Energy Security, Sustainable Development and the Green Bond Market. Energies 2022, 15(17), 6218

Thirdly, there are some technical issues that need to be improved in the paper. The case of the letters denoting the variables in subsection 4.1 should be reduced. In addition, captions should be introduced under tables 1 and 2.

Fourthly, in the model setting the authors proposed regression model trying to examine “the impact of green finance on GTFP” (p. 7). However, the linear regression model is unable to show any impact understood as cause-and-effect relationship. For this purpose, VAR models (in all their varieties) or VEC models are used, which I encourage the authors of the article to use for the purpose of the paper.

Summing up, I recommend to publish the article but after improvements suggested above.

Round 2

Reviewer 2 Report

Comments and Suggestions for Authors

The authors understood their mistakes and corrected them. The authors' explanations are satisfactory. I suggest publishing the peer-reviewed article.

Reviewer 3 Report

Comments and Suggestions for Authors

The authors have been addressed the comments.